# Identification and Analysis of Fungal-Specific Regions in the *Aspergillus fumigatus* Cu Exporter CrpA That Are Essential for Cu Resistance but Not for Virulence

**DOI:** 10.3390/ijms24043705

**Published:** 2023-02-13

**Authors:** Hila Werner, Ammar Abou Kandil, Zohar Meir, Yehonathan Malis, Yona Shadkchan, Gal Masrati, Nir Ben-Tal, Koret Hirschberg, Nir Osherov

**Affiliations:** 1Department of Clinical Microbiology and Immunology, Sackler School of Medicine, Tel-Aviv University, Ramat-Aviv, Tel-Aviv 69978, Israel; 2Department of Pathology, Sackler School of Medicine, Tel-Aviv University, Tel-Aviv 69978, Israel; 3Department of Biochemistry and Molecular Biology, Faculty of Life Sciences, Tel-Aviv University, Tel-Aviv 69978, Israel

**Keywords:** *Aspergillus fumigatus*, Cu export, Cu resistance, Cu^+^ P-type ATPase, protein trafficking, virulence

## Abstract

The opportunistic fungus *Aspergillus fumigatus* is the primary invasive mold pathogen in humans, and is responsible for an estimated 200,000 yearly deaths worldwide. Most fatalities occur in immunocompromised patients who lack the cellular and humoral defenses necessary to halt the pathogen’s advance, primarily in the lungs. One of the cellular responses used by macrophages to counteract fungal infection is the accumulation of high phagolysosomal Cu levels to destroy ingested pathogens. *A. fumigatus* responds by activating high expression levels of *crpA*, which encodes a Cu^+^ P-type ATPase that actively transports excess Cu from the cytoplasm to the extracellular environment. In this study, we used a bioinformatics approach to identify two fungal-unique regions in CrpA that we studied by deletion/replacement, subcellular localization, Cu sensitivity in vitro, killing by mouse alveolar macrophages, and virulence in a mouse model of invasive pulmonary aspergillosis. Deletion of CrpA fungal-unique amino acids 1–211 containing two N-terminal Cu-binding sites, moderately increased Cu-sensitivity but did not affect expression or localization to the endoplasmic reticulum (ER) and cell surface. Replacement of CrpA fungal-unique amino acids 542–556 consisting of an intracellular loop between the second and third transmembrane helices resulted in ER retention of the protein and strongly increased Cu-sensitivity. Deleting CrpA N-terminal amino acids 1–211 or replacing amino acids 542–556 also increased sensitivity to killing by mouse alveolar macrophages. Surprisingly, the two mutations did not affect virulence in a mouse model of infection, suggesting that even weak Cu-efflux activity by mutated CrpA preserves fungal virulence.

## 1. Introduction

Copper (Cu) is a transition metal essential for all living systems from bacteria to eukaryotes. Cu ions serve as important catalytic cofactors in redox chemistry for proteins that are required for growth and development. Cu-requiring proteins are involved in a variety of biological processes such as respiration, protection against oxidative stress, pigment formation, neuro-transmitter biosynthesis, peptide amidation, iron transport, and connective tissue maturation [1]. Cu has an ability to redox cycle between two forms: the extracellular oxidized form (Cu^2+^) and the intracellular reduced form (Cu^+^) and that is the reason for its usefulness as a cofactor. However, the Cu^+^ ↔ Cu^2+^ reversible transition can also transform copper into a toxic compound through a Fenton-like reaction which generates hydroxyl radicals by reacting with hydrogen peroxide, a natural by-product of aerobic respiration [2]. These hydroxyl radicals damage DNA by inducing strand breaks, oxidizing nucleoside bases and inactivating iron–sulfur cluster-containing enzymes, a process shared with hydrogen peroxide [3]. Cu can also cause mismetallation, where it replaces metal cofactors in proteins, rendering them inactive [4]. Because of this dual role, copper being at the same time essential and toxic, all living organisms have developed mechanisms to accurately tune its homeostasis [5]. Cu homeostasis in fungi is mediated by the transcriptional regulation of genes participating in Cu acquisition, mobilization, and sequestration [1]. In response to high toxic extracellular Cu concentrations, fungal transcription factors regulate the expression of genes responsible for the sequestration and efflux of excess Cu [6]. In humans, infection leads to increased concentrations of Cu in the serum, and macrophages accumulate high phagolysosomal Cu levels to destroy ingested pathogens [7,8,9,10].

The Cu-buffering system in the human pathogenic mold *Aspergillus fumigatus* relies primarily on AceA-dependent transcriptional activation of CrpA, encoding a Cu^+^ P-type ATPase highly conserved in eukaryotes and prokaryotes [5,9,11,12,13]. The AceA–CrpA axis is conserved in additional fungi, including *Aspergillus nidulans* [14], *Fusarium oxysporum* [15], and *Candida albicans* [16]. CrpA actively transports excess Cu from the cytoplasm to the extracellular environment. CrpA contains eight transmembrane domains, a conserved CPC (Cys–Pro–Cys) Cu translocation motif in the sixth transmembrane segment and cysteine-rich metal-binding motifs in the cytoplasmic N-terminal, and is apparently localized at the cell surface [11]. Deletion mutants of *aceA* or *crpA* in *A. fumigatus* are hypersensitive to Cu in-vitro. They accumulate higher intracellular Cu levels and are more susceptible to killing by macrophages. In a mouse infection model of invasive pulmonary aspergillosis, these mutants display reduced growth and virulence [9,11]. CrpA overexpression in the *aceA* null background re-establishes a wild-type phenotype, confirming that CrpA is the major effector target gene of AceA [9].

This study aimed at finding fungal-unique domains in the *A. fumigatus* Cu exporter CrpA, and analyze their function. We used a bioinformatics approach to identify two CrpA fungal-specific regions that we studied by deletion/replacement, subcellular localization, Cu sensitivity in vitro, killing by mouse alveolar macrophages, and virulence in a mouse infection model. Deleting CrpA amino acids 1–211 moderately increased Cu sensitivity without altering the localization of the protein. Deleting CrpA amino acids 542–556 strongly increased Cu sensitivity and retention of the protein to the ER. Surprisingly, deleting CrpA amino acids 1–211 or replacement of amino acids 542–556 did not affect *A. fumigatus* virulence in a mouse model of infection, suggesting a more complex role for the involvement of this Cu transporter in the progression of invasive pulmonary aspergillosis.

## 2. Materials and Methods

**Strains and media**. *A. fumigatus* CEA17 KU80 [17], was used to generate the strains described in this study [17]. This commonly used laboratory strain is derived from a patient isolate and has been engineered to be deficient for non-homologous end joining. It therefore has high rates of homologous recombination, making it ideal for genetic manipulation. For routine culture, strains were grown on yeast-extract-rich solid medium (YAG) containing 0.5% (*w*/*v*) yeast extract, 1% (*w*/*v*) glucose, 10 mM MgSO_4_, supplemented with 0.1% (*v*/*v*) trace elements solution, and 0.2% (*v*/*v*) vitamin mix. After incubation for 48 h at 37 °C, conidia were harvested in 0.02% (*v*/*v*) Tween-20, resuspended in double-distilled water (DDW), and counted with a hemocytometer. Conidial stocks were stored at 4 °C for no longer than two weeks. For experiments, strains were grown on defined minimal medium and vitamins (MMV), containing 70 mM NaNO_3_, 1% (*w*/*v*) glucose, 12 mM potassium phosphate pH 6.8, 4 mM MgSO_4_, supplemented with vitamins, trace elements, and 0.1% (*w*/*v*) uracil/uridine (UU) as needed. The strains used in this study and their construction are described in Appendix A.

**Structural and bioinformatic analysis of *A. fumigatus* CrpA**. A model structure of *A. fumigatus* CrpA was built using AlphaFold [18] with templates searched against the pdb70 database. An initial evaluation based on AlphaFold’s PAE score indicated that the position of the first ~285 amino acids relative to the rest of the protein could not be determined at sufficient predicted accuracy. Thus, we divided the protein sequence into N- and C-terminal parts, consisting of residues 1 through 285, and 286 through 1254, respectively, and used AlphaFold to construct model structures of each part separately. Next, we performed a ConSurf evolutionary conservation analysis [18] with the two models. Homologs were collected using HMMER [19] search against the UniRef90 database with E-values of 1 × 10^−5^ and sequence identity ranging from 35% to 95%. Sixty-five homologs were detected for the N-terminal part, and five hundred for the C-terminal part. All detected homologs were of fungal origin. Thus, to identify sequence regions that are unique to fungi and are not found in other eukaryotes, a more thorough homologue search was conducted. Specifically, homologs were searched and aligned using HMMER against all eukaryotic genomes in the UniProt representative proteomes database for both the N- and C-terminal domains, with a 15% co-membership threshold (RP15), an E-value of 1 × 10^−5^, and no constraints on sequence identity. Using that search, 1713 homologs were collected for the N-terminal domain, and 9510 homologs for the C-terminal domain. The RP15 database was chosen to reduce the number of sequences for the initial search. Finally, regions that emerged as potential fungal-specific regions were subjected to a new HMMER search against the full UniProt KB database.

**Microscopy**. CrpA_dN, CrpA_dmid, and CrpA_cont strains were grown on coverslips in MMVUU without copper in 24-well plates for 16 h. The CrpA_cont strain was exposed to different concentrations of copper (0.25, 2.5, 25, and 100 µM) for various time exposures after 16 h of growth. In subsequent experiments, CrpA_dN, CrpA_dmid, and CrpA_cont strains were exposed to 2.5 µM Cu for 2 h. The coverslips were then washed twice with DDW and treated accordingly. Co-localization of the GFP-tagged CrpA protein with the endoplasmic reticulum (ER) or nuclei, was performed with ER-Tracker Red (Invitrogen) or DAPI (4′,6-diamidino-2-phenylindole) (Sigma-Aldrich, St. Louis, MO, USA), respectively. Nuclear staining was carried out for 20 min at room temperature using DAPI solution (50 mM KPO4 pH 6.8, 0.2% (*v*/*v*) Triton X-100, 5% (*v*/*v*) glutaraldehyde, 0.05 µg/mL DAPI). ER staining was performed with 1 µM ER-Tracker Red for 15 min at 37 °C. Microscopy was performed on a Zeiss LSM 800 confocal laser scanning microscope (Zeiss, Jena, Germany).

**Droplet-growth-inhibition assay**. Freshly harvested conidia were serially diluted in sterile water to obtain defined concentrations of 10^6^, 10^5^, 10^4^, and 10^3^ conidia/mL. Conidia were spotted in a volume of 10 µL on MMVUU plates in the presence of increasing concentrations of Cu. Growth was documented after 72 h of incubation at 37 °C.

**Liquid broth inhibition assay**. Freshly harvested conidia (5000 conidia/well), suspended in 200 µL/well MMVUU liquid medium were dispensed in 96-well plates, with increasing concentrations of Cu. The minimal inhibitory concentration (MIC), namely the Cu concentration in which no microscopic growth was visible, was measured after 24 h of incubation at 37 °C.

**Mouse alveolar macrophage conidial killing assay**. Mouse alveolar macrophages were collected from 8-week-old ICR mice as previously described [20]. Conidia (2 × 10^5^) were added to freshly harvested murine alveolar macrophages in 24-well plates at a 1:1 ratio for 2 h of preincubation at 37 °C under 5% CO_2_ in DMEM supplemented with 10% heat-inactivated FBS and 1% penicillin/streptomycin medium. Uninternalized conidia were aspirated and the wells were washed with PBS three times. Then, 1 mL RPMI medium was added for 4 h of incubation at 37 °C and 5% CO_2_. Conidial killing was terminated by aspiration of the RPMI medium and addition of DDW with 0.1% Triton X-100 to lyse the macrophages. Following vigorous scraping of the wells, lysate dilutions were plated on YAG agar plates with 0.01% chloramphenicol and incubated for 24 h at 37 °C to detect viable fungal colonies. Viable counts were compared to those of conidia incubated with macrophages at the 0 h time point (after the 2 h preincubation). The results were analyzed using the Brown–Forsythe ANOVA test with Dunnett’s T3 multiple comparisons post-test in GraphPad Prism software.

**Mouse model of *A. fumigatus* infection**. A standard model of mouse infection was used [21]. Six-week-old ICR female mice were injected subcutaneously with 300 mg/kg cortisone–acetate 3 days before infection, on the day of infection, and 2 and 4 days post-infection to induce an immunocompromised state but without neutropenia. The mice were infected intranasally with 5 × 10^5^ dormant conidia/mouse, suspended in 20 µL of 0.2% Tween 20 in saline (10 µL in each nostril). Survival was monitored for 14 days. The results were analyzed using the log-rank test for Kaplan–Meyer survival curves in GraphPad Prism software.

**Statistical analysis**. Data and statistical analysis were analyzed using GraphPad Prism 5 software package (GraphPad Software, Inc., San Diego, CA, USA) or Microsoft Excel software package (Microsoft Corporation, Redmond, WA, USA). Student’s *t*-test was used for significance testing of two groups. Differences between the groups were considered significant at *p* ≤ 0.05.

## 3. Results

### Bioinformatic Analysis of A. fumigatus CrpA Identifies Two Regions Unique to Fungi

This study aimed at identifying essential fungal-unique regions in fungal CrpA proteins that could serve as therapeutic targets (Figure 1). Figure 1A displays a schematic topological 2-D representation of *A. fumigatus* CrpA, including eight transmembrane domains and the fungal-unique regions identified below, highlighted in red. To analyze the CrpA protein bioinformatically for fungal-unique motifs, we used AlphaFold to construct model structures of *A. fumigatus* CrpA N- and C-terminal regions, consisting of residues 1 through 285 (Figure 1B), and 286 through 1254 (Figure 1C), respectively. The C-terminal region is the main part, which includes the eight transmembrane domains. The figure displays its predicted membrane orientation, inferred from that of a close homologue of known structure, the bacterial *Legionella pneumophila* CopA (PDB entry 4BBJ).

To identify fungal-unique sequences, we performed two multiple sequence alignments for the CrpA N- and C-terminal regions with 1713 and 9510 eukaryotic homologs. Consequently, we identified two fungal-unique protein segments in CrpA. The first is located in the N-terminal cytosolic tail between residues 90 to 200 (Figure 1B, arrow), not including the two short Cu-binding motifs (CSSC, CDAC). Using HMMER [19] and searching against the UniProt KB database specifically for this 110 amino acid section, only 11 other sequences, all from the *Aspergillus* genus, emerged as significant homologs to this N-terminal region. On the basis of AlphaFold’s model, this region is mainly unstructured, aside from two short helices between residues 121 and 130 and residues 135 and 150. The structure is colored on the basis of amino acid position conservation among a collection of homologous fungal proteins, with cyan-through-maroon representing variable-through-conserved, respectively (Figure 1B).

The second fungal-unique segment is located in the C-terminal part of the protein (amino acid 286 through 1254). It consists of an intracellular loop between the second and third transmembrane helices (Figure 1C,D, arrow). Specifically, searching for the sequence corresponding to the second and third helices and their connecting loop resulted in roughly 1030 homologs, all except two are from fungal sequences.

**The CrpA fungal-specific intracellular loop is essential for growth at high concentrations of Cu.** To test whether the fungal-specific N-terminal cytosolic tail and the intracellular loop of CrpA have an essential function in fungi, we constructed two *A. fumigatus* mutant strains (for details, see the Appendix A): (i) CrpA_dN, expressing CrpA without amino acids 1–211. GFP was fused to the CrpA C-terminus to allow spatial localization and expression studies. (ii) CrpA_dmid, expressing CrpA with a replacement of amino acids 542–556 with a flexible glycine–serine linker, and GFP fused to the C-terminus. Both constructs were integrated into the *crpA* deletion site in the Δ*crpA* strain under the endogenous *crpA* promoter. We also constructed a control strain, CrpA_cont, in which wild-type C-terminal GFP-tagged *crpA* was integrated into the *crpA* deletion site in the Δ*crpA* strain as described above.

We analyzed the growth of the three strains (CrpA_dN, CrpA_dmid, and CrpA_cont) (Figure 2) on MMVUU agar plates containing increasing concentrations of Cu (Figure 2A) and in MMVUU liquid broth to determine the minimal inhibitory concentration of Cu (Figure 2B). The deletion mutant Δ*crpA* displayed Cu hypersensitivity (MIC = 2.5 µM). CrpA_dmid was very sensitive to high concentrations of Cu (MIC = 20 µM), indicating that the conserved motif (amino acids 542–556) is important for growth at high concentrations of Cu. The CrpA_dN strain was two-fold more sensitive to high Cu (MIC = 80 µM) than the CrpA_cont strain (MIC = 160 µM), implying that other Cu-binding motifs in the remaining N-terminus of CrpA_dN can be used to bind Cu and thus partially allow Cu export.

**The CrpA intracellular loop is necessary for localizing CrpA to the cell surface.** The reduced Cu resistance of the CrpA_dN and CrpA_dmid strains can be attributed to either CrpA protein instability, mislocalization, or reduced catalytic activity. To determine which of these possibilities is correct, we first examined the expression and subcellular localization of wild-type *A. fumigatus* CrpA, as it has not been previously analyzed in detail (Figure 3). The CrpA_cont strain was grown on coverslips with MMVUU medium without Cu in 24-well plates for 16 h and exposed to different concentrations of Cu for various time exposures (Figure 3A). CrpA-GFP expression was induced by 2.5 µM Cu after 2 h exposure. It did not further increase at later time points (not shown). Exposure to 25 µM Cu did not induce CrpA-GFP expression, possibly due to toxicity. To define the subcellular localization of CrpA, CrpA_cont was exposed to 2.5 µM Cu for 2 h, stained with either DAPI nuclear stain or ER-Tracker Red, and visualized by confocal microscopy. We saw that GFP-tagged CrpA surrounded the nuclei, suggesting ER localization, and was detected in both the intracellular and cell surface membranes (Figure 3B, top panel). CrpA-GFP co-localized with ER-Tracker Red but was also localized at the cell surface (Figure 3B, lower panel), demonstrating that CrpA is transported through the ER onto the cell surface upon Cu induction.

We next compared the expression level and localization of each of the three GFP-tagged strains (CrpA_dN, CrpA_dmid, and CrpA_cont) after exposure to 2.5 µM Cu for 2 h (Figure 4). All strains expressed CrpA-GFP at a similar level, indicating that the mutant proteins were stable. CrpA_dN-GFP and CrpA_cont-GFP localization was very similar and localized to the ER and the cell surface (Figure 4A). However, CrpA_dmid-GFP displayed a strikingly different localization pattern. The CrpA_dmid protein was concentrated in vesicles within the hyphae and not localized to the cell surface. To determine the subcellular localization of CrpA_dmid-GFP, the CrpA_dmid strain was stained with ER-Tracker Red. CrpA_dmid-GFP co-localized to the ER-Tracker Red staining (Figure 4B). These results reveal that the conserved motif (amino acids 542–556) is necessary for CrpA localization to the cell surface. Inability to localize there leads to high Cu sensitivity.

**Deletion of the CrpA N-terminal or intracellular loop increases sensitivity to killing by mouse alveolar macrophages.** We tested if deleting the CrpA N-terminal region or replacing amino acids 542–556 increases sensitivity to killing by mouse alveolar macrophages by co-incubating the cells with conidia at a 1:1 ratio for 2 h to allow conidial internalization, washing away un-internalized conidia, further incubating for 4 h to allow killing of internalized conidia, then lysing the macrophages and plating the conidia on YAG agar plates to calculate the percentage of surviving internalized conidia (Figure 5). Approximately 22% of internalized CrpA_cont conidia survived, compared to 13.1% and 8.4% of CrpA_dN and CrpA_dmid conidia, respectively (*p* = 0.052; *p* = 0.004), while only 1.4% of internalized ΔCrpA conidia survived (*p* < 0.0001) (Figure 5A). The increased susceptibility of the mutant conidia to macrophage killing concurs with their increased Cu sensitivity in vitro, shown above.

**Deletion of the CrpA N-terminal or intracellular loop does not affect virulence in infected mice.** We tested if deleting CrpA N-terminal (amino acids 1–211) or intracellular loop (amino acids 542–556) attenuates virulence in vivo. Because the CrpA_dN, CrpA_dmid, and CrpA_cont described above are uracil auxotrophs, rendering them intrinsically avirulent, we first complemented their deficiency by transformation with AMA-1 *pyr4*, generating uracil prototrophic strains CrpA_dN*, CrpA_dmid*, and CrpA_cont* (see Appendix A for details). Mice (n = 6/group) were immunocompromised and infected intranasally with conidia collected from each of the three strains. We followed mouse survival for 14 days (Figure 5B). Surprisingly, the survival curve showed no difference in virulence between CrpA_dN*-, CrpA_dmid*-, and CrpA_cont*-infected mice, whereas ΔCrpA, in which CrpA is deleted, was reduced in virulence compared with CrpA_cont* as we showed previously (*p* = 0.003) [9]. These results demonstrate that despite the importance of the CrpA N-terminus and the intracellular loop in CrpA localization and function in vitro, these fungal-specific domains do not affect virulence in vivo.

## 4. Discussion

Previous work reported two critical proteins in *A. fumigatus* resistance to toxic Cu levels mounted by the host defense system, namely, the Cu-homeostasis transcription factor AceA and the Cu-exporting ATPase CrpA, positively regulated by AceA [6,9,11]. Deletion of *A. fumigatus crpA* results in Cu hypersensitivity, reduced survival in the presence of mouse alveolar macrophages, and significantly decreased virulence [9]. We therefore decided to investigate the function of CrpA in more detail. In particular, we sought to identify fungal-specific motifs necessary for CrpA function that could be considered as drug targets.

We constructed a model structure of CrpA in silico using AlphaFold and identified two fungal-unique sequences. The first is an N-terminal cytosolic tail (amino acids 1–211), found only in the filamentous ascomycetes. Within it, a ~110 amino acid section spanning roughly between positions 90 and 200 can be found only in the genus *Aspergillus*. The second is an intracellular loop containing a unique conserved motif (amino acids 542–556) found only in fungi and not in other eukaryotic organisms.

We generated *A. fumigatus* strains in which the N-terminal cytosolic tail was deleted (CrpA_dN) or the intracellular loop replaced with a flexible glycine–serine linker (CrpA_dmid) [22]. Deleting the N-terminal cytosolic tail resulted in a moderate increase in Cu sensitivity, and localization of the protein was the same as of wild-type CrpA. There are five predicted Cu-binding domains at the N-terminus of *A. fumigatus* CrpA, and two are located closer to the N-terminus and shared by *A. nidulans* and *C. albicans*, followed by three that are conserved in other fungi [14]. The CrpA_dN strain we generated lacks the two Cu-binding domains closest to the N-terminus (CxxC), and its moderate sensitivity to excess Cu can be explained by the existence of the three remaining metal-binding domains located downstream of the part we deleted. We infer that these three metal-binding domains can bind Cu efficiently enough for the CrpA protein to function at an intermediate level compared to that of the control strain.

In contrast, deletion of the CrpA intracellular loop resulted in high Cu sensitivity. Furthermore, CrpA_dmid was concentrated in the ER and unlike normal CrpA, was not observed at the cell surface. This finding indicates that the fungal-specific intracellular loop (CrpA amino acids 542–556) is essential for trafficking or localizing CrpA to the cell surface. Failure to localize there leads to an inability to efflux excess Cu, and to Cu sensitivity only slightly less severe than that of the *crpA* null strain. On the basis of these findings, the intracellular loop could (i) serve as a motif that directly anchors CrpA to the cell surface, consistent with the high occurrence of arginine residues and aromatic amino acids, known to interact favorably with membrane-lipid head-groups [23]. Alternatively, (ii) the intracellular loop could serve as a motif that targets CrpA for loading into vesicles at the trans-Golgi network and transport via the secretory SEC pathway to the cell surface.

Surprisingly, we detected that although CrpA_dmid was almost as sensitive as the CrpA deletion strain to excess Cu in vitro, unlike the latter, it displayed wild-type virulence in infected mice. This finding suggests that even the small increase in protection against Cu seen in CrpA_dmid compared to ΔCrpA, is sufficient to significantly protect it against Cu stress after ingestion by macrophages and neutrophils. This rationale is strengthened by our results showing that internalized conidia of CrpA_dmid are significantly more resistant to killing by mouse alveolar macrophages compared to ΔCrpA (*p* < 0.0001). Importantly, in our infection model, mouse alveolar macrophages are weakened due to cortisone–acetate administration, which is essential for disease progression. As a result, even the residual Cu resistance in CrpA_dmid could be sufficient to overcome the Cu stress induced by these compromised immune cells.

Collectively, the findings described here and in our previous work [9] show that antifungal therapy based solely on inhibiting CrpA Cu efflux, will, for several reasons, be only partially effective. First, CrpA deletion, attenuates but does not completely block *A. fumigatus* virulence, and second, even residual CrpA activity, as in the CrpA_dmid strain, results in wild-type virulence. Therefore, if fungal-specific CrpA-dependent Cu-efflux inhibitors are developed, they will have to be used with additional potentiating or synergistic drugs.

## 5. Conclusions

*Aspergillus fumigatus*, the most common invasive mold pathogen in humans, activates a dedicated Cu efflux transporter, CrpA, in response to high Cu levels in the phagocytic phagolysosome following infection. Deletion of *A. fumigatus crpA* leads to attenuated virulence, highlighting this transporter as a possible drug target. Because CrpA bears high homology to human Cu-efflux transporters, we used a bioinformatics approach to identify two fungal-unique regions in the *A. fumigatus* efflux transporter CrpA: amino acids 1–211 containing two N-terminal Cu-binding sites, and amino acids 542-556 consisting of an intracellular loop between the second and third transmembrane helices. Deletion of amino acids 1–211 in *A. fumigatus*, resulted in moderately increased Cu sensitivity and increased killing by mouse alveolar macrophages. Deletion of amino acids 542–556 resulted in high Cu sensitivity, strongly increased killing by mouse alveolar macrophages and inability to traffic from the ER to the cell surface. Interestingly, both CrpA deletions did not affect fungal virulence in a cortisone–acetate immunocompromised mouse model of infection. This suggests that even residual Cu resistance is sufficient to maintain *A. fumigatus* virulence during infection and that antifungal therapy based solely on inhibiting CrpA Cu efflux, can be only partially effective.

## Figures and Tables

**Figure 1 ijms-24-03705-f001:**
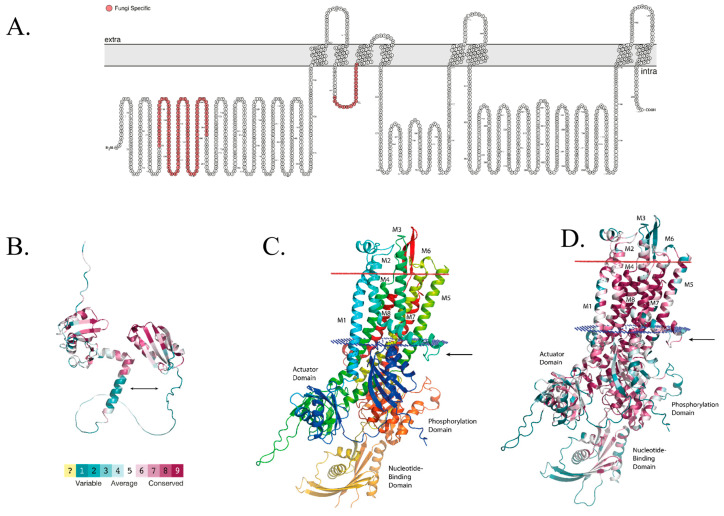
**Model Structure of *A. fumigatus* CrpA.** (**A**) A two-dimensional schematic representation of *A. fumigatus* CrpA topology. Membrane helices are numbered 1 to 8, and the fungal-unique regions CrpA N-terminal domain (amino acids 1 to 285) and intracellular loop (residues 542–556) are highlighted in red. (**B**) A model structure of the *A. fumigatus* CrpA N-terminal domain (amino acids 1 to 285) constructed with AlphaFold and colored on the basis of ConSurf evolutionary conservation analysis with cyan-through-maroon representing variable-through-conserved positions. The unique *Aspergillus* genus region (residues 90–200) is marked with black arrows. (**C**) A model structure of the *A. fumigatus* CrpA C-terminal part (amino acids 285 to 1254) constructed with AlphaFold. The (inferred) extracellular and intracellular leaflets of the membrane are depicted as red and blue discs, respectively. The orientation of CrpA in the membrane was modeled on the basis of the orientation of one of its closest homologs with a known structure, the bacterial *Legionella pneumophila* CopA. The intracellular loop unique to CrpA that connects helices M2 and M3 (residues 542–556) is marked with an arrow. (**D**) A model structure of the *A. fumigatus* CrpA C-terminal region, colored on the basis of ConSurf conservation analysis as in (**B**).

**Figure 2 ijms-24-03705-f002:**
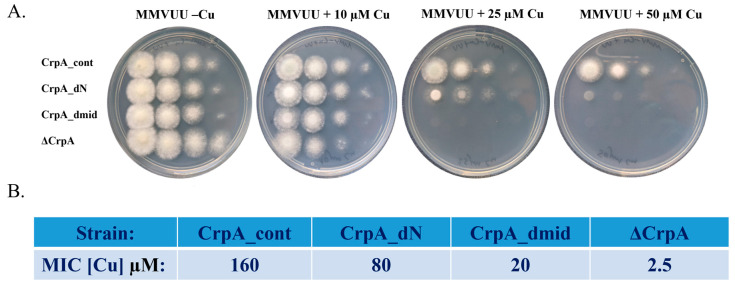
**The *A. fumigatus* CrpA intracellular loop (amino acids 542–556) is essential for growth at high concentrations of Cu.***A. fumigatus* strains CrpA_cont, CrpA_dN (CrpA N-terminal amino acids 1–211 deleted), CrpA_dmid (CrpA intracellular loop amino acids 542–556 replaced), and ΔCrpA strains were grown on MMVUU. (**A**) Agar plates supplemented with increasing concentrations of Cu. Conidial droplets containing increasing numbers of conidia (10, 100, 1000, and 10,000 from right to left) were placed on the plates and incubated for 72 h at 37 °C. (**B**) Liquid broth Cu-sensitivity assay with increasing concentrations of Cu for 48 h at 37 °C for MIC determination.

**Figure 3 ijms-24-03705-f003:**
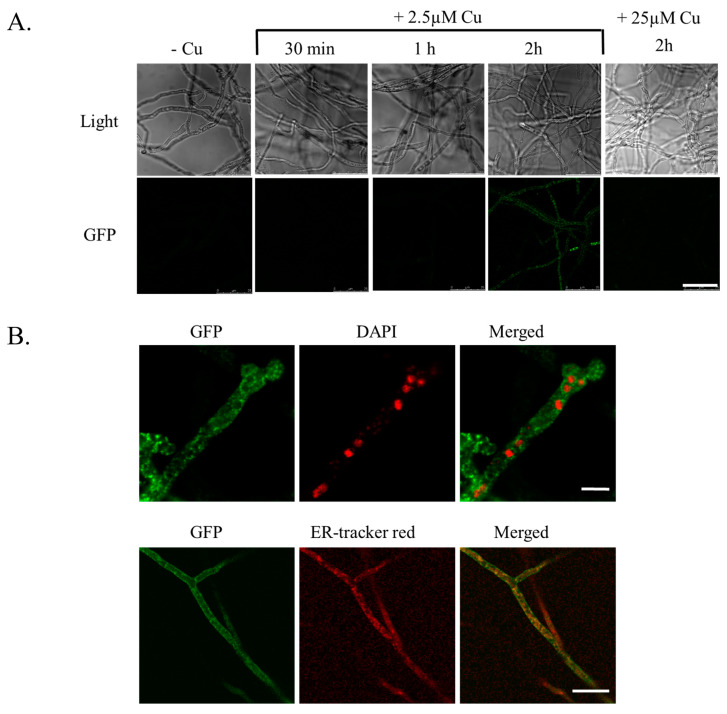
**Cu-inducible expression of *A. fumigatus* CrpA-GFP and localization to the ER.** CrpA_cont conidia were grown in liquid MMVUU for 16 h at 37 °C and then shifted to Cu. (**A**) CrpA-GFP expression after a shift to 2.5 µM or 25 µM Cu for up to 2 h. Size-bar = 25µm. (**B**) CrpA-GFP localization after a shift to 2.5 µM for 2 h and nuclear staining with DAPI (top panel) (size-bar = 2.5 µm) or ER-Tracker Red (lower panel) (size-bar = 10 µm). Fluorescence images were acquired using confocal microscopy.

**Figure 4 ijms-24-03705-f004:**
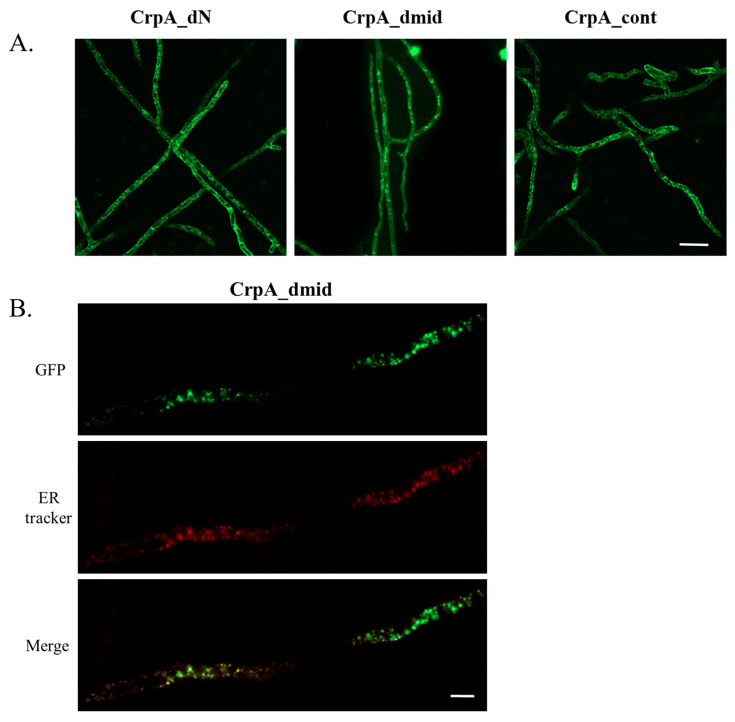
**The CrpA intracellular loop (amino acids 542–556) is essential for the localization of CrpA to the cell surface.** (**A**) Subcellular localization of CrpA_dN-GFP (N-terminal amino acids 1–211 deleted), CrpA_dmid-GFP, (intracellular-loop amino acids 542–556 replaced), and CrpA-GFP full length protein (CrpA_cont). Fluorescence images were acquired using confocal microscopy under identical exposure settings. (**B**) CrpA_dmid-GFP localization (top panel) after Cu shift and staining with ER-Tracker Red (lower panel) (size-bar = 2.5 µm). In both panels (**A**,**B**), conidia were grown on coverslips in liquid MMVUU for 16 h at 37 °C. Following that, 2.5 µM Cu was added for 2 h. Fluorescence images were then acquired using confocal microscopy.

**Figure 5 ijms-24-03705-f005:**
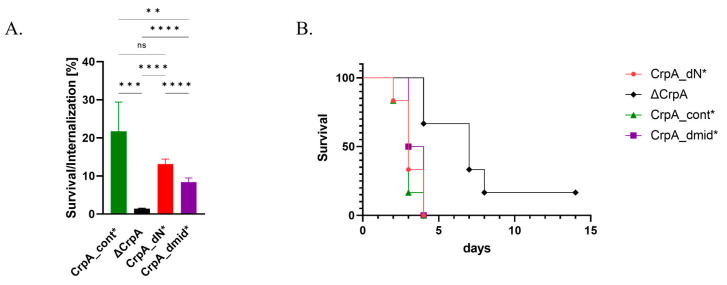
Deletion of CrpA N-terminal amino acids 1–211 or replacement of amino acids 542–556 increases susceptibility to murine alveolar macrophages, but does not affect virulence in infected mice. (**A**) Survival rates of CrpA_cont, CrpA_dN, CrpA_dmid, and ΔCrpA conidia internalized by murine alveolar macrophages for 4 h at 37 °C. (**B**) Mouse survival curves after intranasal infection of cortisone–acetate immunocompromised mice (n = 6 animals/group) with *A. fumigatus* CrpA_cont, CrpA_dN, CrpA_dmid, and ΔCrpA conidia. Results were analyzed using the log-rank test for Kaplan–Meyer survival curves in GraphPad Prism software. ns: *p* > 0.05; *: *p* ≤ 0.05; **: *p* ≤ 0.01; ***: *p* ≤ 0.001; ****: *p* ≤ 0.0001 (for the last two choices only).

## Data Availability

Data is contained within the article or Appendix A.

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
