# Peer review of "Identification and Analysis of Fungal-Specific Regions in the Aspergillus fumigatus Cu Exporter CrpA That Are Essential for Cu Resistance but Not for Virulence"

_ijms, 2023, doi:10.3390/ijms24043705_

Round 1

Reviewer 1 Report

In the manuscript ijms-2062205, the authors identified by a bioinformatics approach 2 fungal-specific regions in the Aspergillus fumigatus Cu exporter CrpA, which they studied by deletion/replacement, subcellular localization, Cu sensitivity in vitro, killing by mouse alveolar macrophages, and virulence in a mouse infection model. They conclude that these regions are essential for Cu resistance but not for virulence.

The following comments/suggestions are offered for the author's consideration:

The subject is fundamentally interesting and could lead to the discovery of new antifungal targets. The authors have deployed an enormous amount of work, both for the construction of the strains, the phenotypic tests and the in vivo tests on a mouse model.

The manuscript is well constructed and well written. References are relevant, methods and results are clearly explained and the discussion is argued and balanced.

It is a work of very good quality on the scientific level, but the conclusions of the confocal microscopy are not argued enough in my opinion to accept the paper without revision:

Lines 210-228: I find it difficult to share the authors' conclusions on the subcellular localization of CrpA from figure 3B: on the top panel, it is clearly seen that the GFP and DAPI fluorescences do not overlap, indicating that CrpA is not located at the nuclear level. But on the bottom panel, I don't see how the authors can conclude definitely on plasma membrane localization. In the merged figure, the green border around is not necessarily synonymous with membrane localization. It could correspond to a submembrane localization. Furthermore in fungi, there is also a parietal compartment. To make the distinction between the two, it would be necessary to repeat the experiment by adding fluorochromes specifically localizing on fungal membranes or cell walls (example: calcofluor white...).

Lines 229-240: same conclusions for figure 4. Figure 4A cannot demonstrate localization to endoplasmic reticulum and plasma membrane by GFP fluorescence alone. As for figure 4B concerning the CrpA_dmid-GFP strain, the confocal microscopy only makes it possible to conclude that the cytoplasmic localization at the endoplasmic reticulum of CrpA is heterogeneous, but not that the deleted motif is necessary for the plasma membrane localization of the CrpA protein. Other fluorochromes are needed to reach this conclusion.

In addition, I have 2 questions, which the authors could answer quickly, either in "material and methods" or in the discussion:

Why did you use the CEA17 KU80 strain to generate the transformed strains in this study?

Why did you use ICR mice, both for mouse alveolar macrophage conidial killing assay and for mouse model of fungal infection?

Author Response

The manuscript is well constructed and well written. References are relevant, methods and results are clearly explained and the discussion is argued and balanced.

It is a work of very good quality on the scientific level, but the conclusions of the confocal microscopy are not argued enough in my opinion to accept the paper without revision:

WE WOULD LIKE TO THANK THE REVIEWER FOR HIS NICE WORDS AND ATTENTIVE CORRECTIONS AND FEEDBACK

Lines 210-228: I find it difficult to share the authors' conclusions on the subcellular localization of CrpA from figure 3B: on the top panel, it is clearly seen that the GFP and DAPI fluorescences do not overlap, indicating that CrpA is not located at the nuclear level. But on the bottom panel, I don't see how the authors can conclude definitely on plasma membrane localization. In the merged figure, the green border around is not necessarily synonymous with membrane localization. It could correspond to a submembrane localization. Furthermore in fungi, there is also a parietal compartment. To make the distinction between the two, it would be necessary to repeat the experiment by adding fluorochromes specifically localizing on fungal membranes or cell walls (example: calcofluor white...).

REPLY- The term “plasma membrane” has been replaced with “cell surface”. A similar terminology was used in ref. 14 to describe Aspergillus nidulans CrpA-GFP localization in the absence of another membrane-specific co-stain (Ansotegi-Uskola et. al.)

Lines 229-240: same conclusions for figure 4. Figure 4A cannot demonstrate localization to endoplasmic reticulum and plasma membrane by GFP fluorescence alone. As for figure 4B concerning the CrpA_dmid-GFP strain, the confocal microscopy only makes it possible to conclude that the cytoplasmic localization at the endoplasmic reticulum of CrpA is heterogeneous, but not that the deleted motif is necessary for the plasma membrane localization of the CrpA protein. Other fluorochromes are needed to reach this conclusion.

REPLY- The term “plasma membrane” has been replaced with “cell surface”. A similar terminology was used in ref. 14 to describe Aspergillus nidulans CrpA-GFP localization in the absence of another membrane-specific co-stain (Ansotegi-Uskola et. al.)

In addition, I have 2 questions, which the authors could answer quickly, either in "material and methods" or in the discussion:

Why did you use the CEA17 KU80 strain to generate the transformed strains in this study?

REPLY- This has been explained in the Methods, L101-104

Why did you use ICR mice, both for mouse alveolar macrophage conidial killing assay and for mouse model of fungal infection?

REPLY- ICR mice are the standard model for modeling invasive aspergillosis. We used the same strain for both the macrophage conidial killing assay and the infection experiments because this enables us to directly compare between them. This has been added in the Methods L173-4

Reviewer 2 Report

The authors must write at the end of the introduction the objective of the study, by example: This study aimed to analysis of fungal-specific regions in the Aspergillus fumigatus Cu exporter CrpA. The manuscript lacks conclusions as a separate chapter after discussion. The text of the manuscript is incomplete. Update references, by example: Anabosi D, Meir Z, Shadkchan Y, Handelman M, Abou-Kandil A, Yap A, Urlings D, Gold MS, Krappmann S, Haas H, Osherov N. Transcriptional response of Aspergillus fumigatus to copper and the role of the Cu chaperones. Virulence. 2021 Dec;12(1):2186-2200. doi: 10.1080/21505594.2021.1958057.

Author Response

WE THANK THE REVIEWER FOR HIS IMPORTANT FEEDBACKS

The authors must write at the end of the introduction the objective of the study, by example: This study aimed to analysis of fungal-specific regions in the Aspergillus fumigatus Cu exporter CrpA.

REPLY- This has been added, L88-89.

The manuscript lacks conclusions as a separate chapter after discussion.

REPLY- A conclusion has been added (L354-371)

 The text of the manuscript is incomplete. Update references, by example: Anabosi D, Meir Z, Shadkchan Y, Handelman M, Abou-Kandil A, Yap A, Urlings D, Gold MS, Krappmann S, Haas H, Osherov N. Transcriptional response of Aspergillus fumigatus to copper and the role of the Cu chaperones. Virulence. 2021 Dec;12(1):2186-2200. doi: 10.1080/21505594.2021.1958057.

REPLY- This reference has been updated (L75)

Reviewer 3 Report

 The article entitled Identification and analysis of fungal-specific regions in the Aspergillus fumigatus Cu exporter CrpA that are essential for Cu resistance but not for virulence presents the bioinformatics approach to identify two fungal-unique regions in CrpA that we studied by deletion/replacement, subcellular localization, Cu sensitivity in vitro, killing by mouse alveolar macrophages, and virulence in a mouse model of invasive pulmonary aspergillosis. Further, the authors demonstrated that deleting CrpA N-terminal amino acids 1-211 or replacing amino acids 542-556 also increased sensitivity to killing by mouse alveolar macrophages. Although the approach suits the scope of the journal and presents a systematic investigation,  the article requires some minor revisions to be publishable in IJMS.

1. I suggest elaborating the introduction with some background discussions.

2. Improve the subsection headings with short and concise titles.

3. Suggest citing more refs from recently published related works.

Suggest, quantifying the data in Figure 2.

4. Separate the conclusion from the Discussion section for better insight

5. Suggest providing the schematic abstract demonstrating the outline of the study

6. The language needs to be improved by a native speaker.

Author Response

WE THANK THE REVIEWER FOR HIS HELPFUL COMMENTS

the article requires some minor revisions to be publishable in IJMS.

  1. I suggest elaborating the introduction with some background discussions.

REPLY- The introduction has been substantially elaborated and references added (L51-68)

  1. Improve the subsection headings with short and concise titles.

REPLY- The subsection headings have been shortened and made more concise (see results section)

  1. Suggest citing more refs from recently published related works

REPLY- Additional refs have been added in L75-77.

Suggest, quantifying the data in Figure 2.

REPLY- This has been added by performing quantitative MICs of the mutants shown in the figure. See the description in the methods (L156-160), the MIC table added to Fig. 3, and description in the methods section (L156-160), results section (L232-238) and figure legend (P19 top).

  1. Separate the conclusion from the Discussion section for better insight

REPLY- A conclusion has been added (L354-371)

  1. Suggest providing the schematic abstract demonstrating the outline of the study

REPLY- A schematic has been added.

  1. The language needs to be improved by a native speaker.

REPLY- A native speaker has gone over the manuscript again and corrected it.